# Similarity-Based Load-Balancing for Distributed Genomic Indices

Woodward Galbraith
Northeastern University
Boston, MA
w.galbraith@northeastern.edu

Ilgar Mammadov
Northeastern University
Boston, MA
mammadov.i@northeastern.edu

Shreyaan Pathak
Northeastern University
Boston, MA
pathak.shr@northeastern.edu

Benjamin M. Gyori
Northeastern University
Boston, MA
b.gyori@northeastern.edu

Prashant Pandey
Northeastern University
Boston, MA
p.pandey@northeastern.edu

## ABSTRACT

Publicly accessible large-scale sequencing data repositories are expanding rapidly. For example, the NIH Sequence Read Archive (SRA) alone now holds over 50 petabases of data and grows 16% larger every year. Enabling efficient search over these vast datasets is key to advancing biological discovery. Recent work has introduced updatable, distributed search indices over unassembled sequencing experiments at petabase scale. However, current methods do not consider sequence similarity when distributing new experiments to nodes, leading to redundant storage and inflated index sizes, a bottleneck for keeping up with ever-expanding modern genomic repositories. To address this, we propose a similarity-based load-balancing strategy, which can be applied to any updatable distributed genomic search index, and implement a system using genomic sketching methods to distribute files across nodes. Our initial results, evaluated on a 2000-experiment subset of the Breast, Blood, and Brain dataset from SRA, demonstrate that similarity-based load-balancing reduces aggregate $k$-mer count by $\approx 8.6\%$, worker $k$-mer variance by $\approx 61\%$, and color class table sizes by $\approx 4.7\%$ and further, that these compactness gains scale with both number of experiments and worker count, compared to naive routing methods, indicating this is a viable path towards more compact and better load-balanced distributed genomic search indices.

**VLDB Workshop Reference Format:**
Woodward Galbraith, Ilgar Mammadov, Shreyaan Pathak, Benjamin M. Gyori, and Prashant Pandey. Similarity-Based Load-Balancing for Distributed Genomic Indices. VLDB 2026 Workshop: Biomedical Data Management Systems (BioDMS).

**VLDB Workshop Artifact Availability:**
The source code, data, and/or other artifacts have been made available at https://github.com/buzgalbraith/mantis_lb.

## 1 INTRODUCTION

Sequencing data capturing the nucleotide composition of DNA or RNA molecules from biological samples is crucial for understanding the molecular basis of health and disease. There is a vast and ever-expanding pool of such data stored in public repositories, for example, the NIH Sequence Read Archive (SRA) [15], which holds more than 50 petabases of sequences and grows by 16% each year. Enabling searching for specific sequences at repository scale holds the potential to transform biological discovery.

However, prior work, such as BLAST [6] and its descendants, are ill-suited for indexing unassembled data. Such data, typically stored as FastQ files encoding sequencing reads and quality, comprise the vast majority of public sequencing data. Efforts such as Mantis [21] and The Sequence Bloom Tree [29] address this by viewing sequences as unordered sets of $k$-mers, re-framing the task from checking substrings to set inclusion. Recent work such as Metagraph [14] and a distributed expansion to the Mantis system, which is currently in development, builds on this by creating distributed search indices over unassembled sequencing data, which support incremental updates as new experiments are added.

As distributed variants of sequence search indices do not consider the structure and composition of the underlying sequences when distributing new experiments to nodes, similar files may be assigned to different sub-indices, causing unnecessary $k$-mer replication across indices and ultimately producing a larger overall index. This creates a problematic bottleneck for supporting space-efficient incremental updates required for constantly expanding modern sequencing repositories.

To address this gap, we propose a similarity-based load-balancing strategy, leveraging genomic sketching methods, such as Mash [20], SourMash [25] and Dashing2 [7], to enable efficient similarity estimation between large sequencing datasets. Furthermore, our load-balancing strategy is independent of the underlying $k$-mer indexing method and can be applied to any distributed, updatable genomic search index. To evaluate this method, we implement a system leveraging SourMash similarity estimation to distribute new files across Mantis indices, and evaluate on a 2000-experiment subset of the Breast, Blood, and Brain RNA-seq benchmark dataset [29]. Our results demonstrate that similarity-based load-balancing reduces the total Mantis color class table size by approximately 4.7%, total $k$-mer count by approximately 8.6% and total $k$-mer variance by approximately 61%, compared to the baseline Round-robin and weighted-random load-balancing strategies, and further that these

improvements to compactness seem to scale with worker count and amount of experiments distributed suggesting that grouping genomically similar experiments is a viable strategy for reducing index redundancy and improving load balance in distributed genomic search systems.

## 2 BACKGROUND

### 2.1 Colored de Bruijn graphs

De Bruijn graphs (DBGs) are widely used to represent the topological structure of a set of $k$-mers [9, 10, 22, 24, 27, 28, 33].

*Definition 2.1.* Given a set $E$ of $k$-mers, the *DBG induced by E* has edge set $E$, where each $k$-mer (or edge) connects its two $(k-1)$-length substrings (or vertices).

Colored de Bruijn graphs (CDBGs) extend the DBG by assigning each edge (or node) $x$ of the DBG a *color class $C(x)$* drawn from some universe $U$. Examples of $U$ and $C(x)$ are:

- Sometimes, $U$ is a set of reference genomes, and $C(x)$ is the subset of those containing $k$-mer $x$ [4, 5, 16, 17].
- Sometimes, $U$ is a set of *reads*, and $C(x)$ is the subset of reads containing $x$ [1, 2, 32].
- Sometimes, $U$ is a set of sequencing experiments, and $C(x)$ is the subset of those containing $x$ [21, 29–31].

The goal of a CDBG representation is to store $E$ and $C$ as compactly as possible[1], while supporting the following operations efficiently:

- *Point query.* Given a $k$-mer $x$, determine whether $x$ is in $E$.
- *Color query.* Given a $k$-mer $x \in E$, return $C(x)$.

Given that we can perform point queries, we can traverse the DBG by simply querying for the 8 possible predecessor/successor edges of an edge. This enables us to implement more advanced algorithms, such as bubble calling [12].

### 2.2 Classic and MST-Based Mantis

Both classic and Minimum Spanning Tree (MST) based Mantis map $k$-mers to the set of samples in which they occur, which we call a $k$-mer's "color". Both versions of Mantis perform this mapping in two steps. In the first step, they map each $k$-mer to a color-class ID. In the second step, each color-class ID is mapped to a color. The first step is performed using Squeakr [23] to construct a counting quotient filter (CQF), which is a compact hash table for small keys and values [22]. Both versions of Mantis ensure that $k$-mers that occur in the same set of samples are mapped to the same color-class ID, so the mapping from color-class ID to color is injective. Furthermore, both versions of Mantis assign lower IDs to popular colors, since color IDs are encoded using variable-length counters in the quotient filter.

Classic and MST-based Mantis differ in how they map color-class IDs to colors. In classic Mantis, color-class IDs are essentially indices into an array of bit-vectors. Each bit-vector in the array encodes a set of samples. The array of bit-vectors is compressed using RRR [26], which supports random access to entries in the array. This is similar to the color encoding in Rainbowfish [4].

---

[1]The nodes of the DBG are typically stored implicitly, because the node set is simply a function of $E$.

MST-based Mantis [3] compresses the array of bit-vectors further by exploiting similarities between colors. MST-based Mantis organizes the color-class IDs into a tree rooted at the all-zeros vector. It then stores, for each node, the node's parent ID and the bitwise difference between the node's bit-vector and its parent's bit-vector. Thus, one can reconstruct the bit-vector for any node in the tree by applying all the bitwise differences along the path from the node to the root. MST-based Mantis constructs the tree by first constructing a weighted graph $C$ on all the color-class IDs. For each edge in the DBG connecting $k$-mers $k_1$ and $k_2$ it adds an edge to $C$, connecting $k_1$ and $k_2$'s color-class IDs. The weight on this edge is the Hamming distance between the IDs' corresponding bit-vectors. MST-based Mantis then computes an MST for this graph. This approach ensures that the total size of the representation of all the bitwise differences in the MST is minimized. This encoding reduces the size of the color class table by an order of magnitude.

### 2.3 Sketching and Similarity

Sketching methods are a class of probabilistic data structures that are widely used for efficient similarity estimation between sequencing experiments [7, 13, 19, 20]. The foundational genomic sketching technique Mash [20] adapts the MinHash [8] locality-sensitive hashing method for sequence comparison. MinHash works by applying a set of hash functions to the elements of a set and retaining only the minimum hash value from each function. The fraction of minimum hash values shared between two sketches then provides an unbiased estimator of their Jaccard similarity. Formally, given the sets of $k$-mers in two sequencing studies $A$ and $B$ we can estimate their Jaccard similarity by

$$J(A, B) = \frac{|A \cap B|}{|A \cup B|} \approx \frac{|S(A \cup B) \cap S(A) \cap S(B)|}{|S(A \cup B)|} \quad (1)$$

where $S(\cdot)$ provides the bottom k hash values of a set's elements, constituting a representative random sample. SourMash expands on this leveraging FracMinHash [13] sketching to retain all $k$-mers under a threshold scaled by experiment size, instead of a fixed bottom $K$ $k$-mers; allowing for unbiased Jaccard estimation for experiments of different size and containment estimation.

## 3 METHODS

### 3.1 Experimental Design

To explore similarity-based methods for worker assignment in updatable distributed genomic search indices, we implemented a system using Mantis as our underlying index, assuming:

(1) We maintained $N$ separate Mantis indices on a single machine to simulate a distributed system.
(2) Node assignment for incoming experiments is performed online, with each experiment routed sequentially according to the load-balancing strategy. However, to reduce computational overhead, Mantis indices were constructed only after all files had been assigned to workers, rather than updated incrementally. This does not affect the validity of the load-balancing comparison, as index size is determined by worker file assignment.

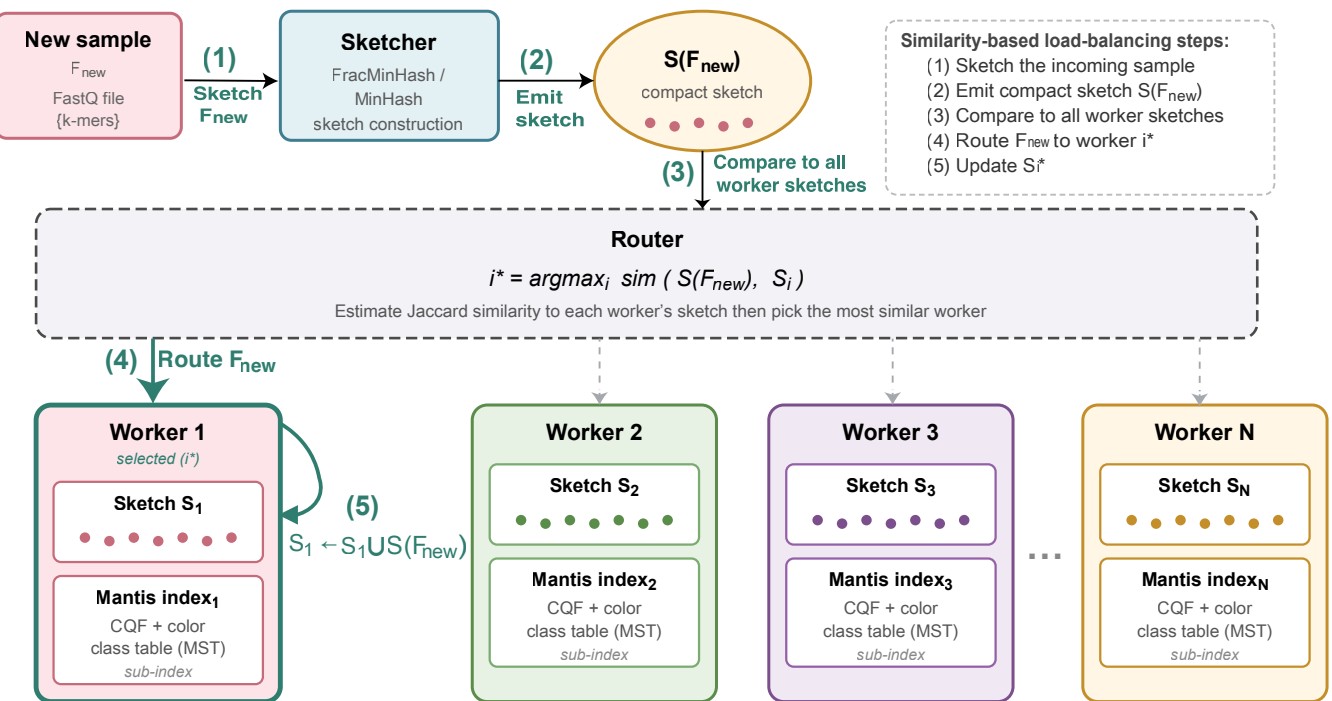

**Figure 1: Similarity-based load-balancing: incoming files are routed to the most genomically similar worker using sketch-based similarity estimation, and the selected worker's sketch is updated accordingly.**

## 3.2 Load-Balancing Strategies

*3.2.1 Round-robin Load-Balancing.* Round-robin is the baseline load-balancing strategy currently deployed in the Amazon Web Services (AWS)-based Mantis distributed system, operating as follows:

(1) Workers are numbered 1 to $N$.
(2) Sequences are assigned to workers in order.
(3) After worker $N$, assignment wraps back to worker 1.

*3.2.2 Similarity-Based Load-Balancing.* The similarity-based strategy, shown in Figure 1, maintains a sketch $S_i$ of the $k$-mers assigned to each worker $i$. For a new input file $F_{new}$, the load-balancer:

(1) Computes a compact sketch $S(F_{new})$.
(2) Emits sketch $S(F_{new})$ to the router.
(3) Compares $S(F_{new})$ against each worker's sketch $S_i$.
(4) Routes $F_{new}$ to worker $i^* = \arg\max_i \text{sim}(S(F_{new}), S_i)$.
(5) Updates $S_{i^*} \leftarrow S_{i^*} \cup S(F_{new})$.

*3.2.3 Weighted-random Load-Balancing.* To verify that any change in Mantis index size stems from grouping genomically similar experiments, rather than the uneven number of files assigned to workers by each method, we introduce weighted-random load-balancing as a randomized control baseline. Instead of assigning files based on genomic similarity, weighted-random draws worker assignments from a weighted distribution, where each worker's weight is proportional to the number of files the similarity-based method assigned to it. For example, if the similarity-based strategy assigned 178 files from a set of 750 files to a worker, the corresponding weighted-random strategy would have a $\frac{178}{750}$ chance of randomly assigning a given file to that worker. If similarity-based load-balancing actually

improves index compactness, we would expect it to outperform both the Round-robin and weighted-random baselines.

## 4 RESULTS

To benchmark our method, we used SRA-tools to pull a 2000-experiment (2,659 FastQ files including paired-end reads) subset of the Breast, Blood, and Brain RNA-seq benchmark dataset [29]. We then used Round-robin assignment to distribute 500 initial experiments to 5 simulated worker nodes each maintaining a separate Mantis index. Following this, we simulated incremental updates to these indices using each load-balancing method to distribute the remaining 1,500 experiments among our simulated workers. Finally, we compared the index sizes, index representation, and runtime of each method.

As in [21], all tools were run with a file-size adaptive $k$-mer cut-off in the CQF and $k$-mer length 20. Further, all experiments were run on a RHEL 9.3 HPC system with 32 CPUs and 64GB of RAM, with an NFS file system. Finally, Mantis index construction was run as a job-array with one job per index.

## 4.1 Index Size Comparison

Table 1 shows the index size of various Mantis outputs on disk across load-balancing strategies. We can see that the CQF size is consistent across methods. This is because the CQF sizes are set by the base-2 logarithm of the number of slots (a power of two), which may not vary given the relatively small change in the number of $k$-mers. The color class table constructed using the similarity-based method is 4.7% smaller than that produced by Round-robin and

| Worker | CQF size (GB) | | | CT size (MB) | | | MST CT size (MB) | | |
|---|---|---|---|---|---|---|---|---|---|
| | RR | WR | Sim | RR | WR | Sim | RR | WR | Sim |
| 1 | 2.78 | 2.78 | 2.78 | 588.4 | 719.4 | 647.2 | 146.7 | 172.2 | 157.4 |
| 2 | 2.78 | 2.78 | 2.78 | 627.3 | 648.1 | 654.2 | 154.9 | 158.3 | 163.2 |
| 3 | 2.78 | 2.78 | 2.78 | 639.8 | 551.5 | 605.3 | 155.1 | 132.9 | 144.7 |
| 4 | 2.78 | 2.78 | 2.78 | 646.7 | 644.2 | 514.6 | 158.1 | 155.9 | 140.8 |
| 5 | 2.78 | 2.78 | 2.78 | 609.2 | 540.9 | 544.1 | 150.0 | 131.3 | 139.1 |
| Total | 13.91 | 13.91 | 13.91 | 3111.4 | 3104.2 | **2965.5** | 764.8 | 750.6 | **745.2** |

**Table 1: Per-worker size of Mantis CQF, color table (CT), and MST color table (MST CT) representations for Round-robin (RR), weighted-random (WR), and similarity-based (Sim) methods, after distributing 1500 experiments across 5 simulated workers initialized with 500 experiments (100 each).**

| Worker | Files | | | $k$-mers ($10^7$) | | | Color Classes ($10^6$) | | |
|---|---|---|---|---|---|---|---|---|---|
| | RR | WR | Sim | RR | WR | Sim | RR | WR | Sim |
| 1 | 533 | 587 | 587 | 16.24 | 17.26 | 15.49 | 19.06 | 21.91 | 20.15 |
| 2 | 533 | 539 | 530 | 14.57 | 16.50 | 14.07 | 20.12 | 20.61 | 21.12 |
| 3 | 531 | 496 | 523 | 13.43 | 13.71 | 12.43 | 20.29 | 18.51 | 19.29 |
| 4 | 531 | 552 | 524 | 18.76 | 16.41 | 15.36 | 20.76 | 20.33 | 17.53 |
| 5 | 531 | 485 | 495 | 15.99 | 15.20 | 14.81 | 19.66 | 18.63 | 18.85 |
| Total | 2659 | 2659 | 2659 | 78.99 | 79.08 | **72.17** | 99.89 | 99.99 | **96.94** |

**Table 2: File, $k$-mer, and color class counts for Round-robin, weighted-random, and similarity-based methods after distributing 1500 experiments across 5 simulated workers initialized with 500 experiments (100 each).**

4.5% smaller than that produced by weighted-random. Similarly, the MST color class table is 2.6% smaller than Round-robin and 0.7% smaller than weighted-random.

## 4.2 Index Representation

Table 2 compares the effect of load-balancing strategy on worker file distribution and Mantis index representation. We can see that the similarity-based strategy reduces the number of redundant $k$-mers stored across worker CQFs compared to naive routing methods, achieving an approximately 8.6% reduction in mean $k$-mer count compared to the Round-robin approach. Additionally, the similarity-based strategy reduces the number of color classes, the unique set of experiments in which a $k$-mer occurs, in the Mantis color class table, on average by approximately 3% over the Round-robin approach. Further, we see a 61% reduction in worker CQF $k$-mer variance with similarity-based over Round-robin assignment, indicating that load is better balanced among workers. Though variance in worker color-class count rises, as CQFs constitute the majority of index size, their variance is a more significant load-balancing metric. Importantly, the weighted-random load-balancing strategy replicates the file-count distribution of the similarity-based method using random assignment, yet still produces larger Mantis representations. This confirms that our method's efficiency gains stem from grouping genomically similar experiments and are not simply the result of distributing fewer files to some workers than the Round-robin approach.

| Task | Runtime |
|---|---|
| Build worker sketches | 5:37:11 ± 0:40:13 |
| Round-robin load balancing | 0:00:23 ± 0:00:01 |
| weighted-random load balancing | 0:00:04 ± 0:00:01 |
| Similarity load balancing | 12:30:01 ± 1:54:35 |
| Squeakr construction | 10:51:31 ± 1:01:31 |
| Mantis cluster construction | 2:45:17 ± 0:23:58 |

**Table 3: Mean wall-clock runtime (± one standard deviation) across 4 repeated trials for each stage of the Mantis index construction pipeline using 500 experiments for the initial index and 1500 for load balancing.**

## 4.3 Runtime Comparison

Table 3 shows the distribution of wall-clock runtime for each stage of the load-balancing and index construction pipeline across 4 repeated trials. The total cost of the pipeline for the baseline load-balancing methods is the method's runtime plus Squeakr and Mantis construction time. The similarity-based method additionally requires the one-time construction of worker sketches, which are used for worker assignment during load-balancing. We can see that the similarity-based method is substantially more costly than the Round-robin strategy, more than doubling total runtime, which likely outweighs the potential space savings gained from similarity-based routing. This is likely caused by the naive parallelism utilized in our current sketching method which assigns each thread to independently sketch a given FastQ file, quickly saturating available I/O bandwidth and limiting the benefit of additional threads. This is a limitation of our prototype system that we plan to address in subsequent work. Additionally, we observed that variance in runtime seems to scale with both total duration and resource intensity of each pipeline stage. We attribute this to node heterogeneity on the shared HPC cluster used for testing, rather than the algorithm itself. Note that as our system is distributing experiments to indices on a single machine, reported runtime numbers do not account for network latency present in a true distributed system.

## 4.4 Scaling Results

To assess whether the benefits of similarity-based load-balancing persist beyond our initial evaluation, we examined how relative performance compares to baseline strategies as we vary the number of experiments distributed and the number of workers available. Note that all runs, as in the previous section, distributed 500 experiments among worker nodes to construct initial Mantis indexes with Round-robin assignment prior to load-balancing. Table 4 reports percent change in index size and representation as the number of experiments to distribute increases from 250 to 1500, with worker count fixed at 5. We can see that the relative performance gains from similarity-based load-balancing over baseline strategies increase as the number of experiments distributed grows for a fixed number of workers. For example, the reduction in aggregate $k$-mer count relative to Round-robin grows from $\approx 0.5\%$ while distributing 250 experiments to $\approx 8.6\%$ for 1500. We attribute this to the growing likelihood of highly similar experiments appearing in larger datasets, which similarity-based routing can increasingly exploit to reduce redundancy. Table 5 reports the same metrics as worker count increases from 3 to 9, with experiment count fixed at 1500. We

| # Experiments | CQF % $\Delta$ | | CT % $\Delta$ | | MST CT % $\Delta$ | |
|---|---|---|---|---|---|---|
| | RR | WR | RR | WR | RR | WR |
| 250 | 0.00% | 0.00% | -1.38% | -0.37% | -0.50% | 0.26% |
| 500 | 0.00% | 0.00% | -2.30% | -1.26% | -1.10% | 0.56% |
| 1000 | 0.00% | 0.00% | -3.02% | -2.56% | -0.69% | -0.67% |
| 1500 | 0.00% | 0.00% | -4.69% | -4.47% | -2.56% | -0.72% |

(a) Index size.

| # Experiments | $k$-mer % $\Delta$ | | Color Classes % $\Delta$ | |
|---|---|---|---|---|
| | RR | WR | RR | WR |
| 250 | -0.58% | -1.11% | -0.62% | 0.42% |
| 500 | -3.42% | -3.04% | -1.67% | -0.55% |
| 1000 | -4.71% | -5.06% | -1.96% | -1.82% |
| 1500 | -8.64% | -8.74% | -2.95% | -3.05% |

(b) Index representation.

**Table 4: Percent change in index size and representation for similarity-based load-balancing relative to Round-robin (RR) and weighted-random (WR) baselines, as the number of experiments distributed increases, for a system with 5 simulated workers, each initialized with 100 of 500 total experiments via Round-robin assignment.**

| # Workers | CQF % $\Delta$ | | CT % $\Delta$ | | MST CT % $\Delta$ | |
|---|---|---|---|---|---|---|
| | RR | WR | RR | WR | RR | WR |
| 3 | 0.00% | 0.00% | -3.39% | -2.99% | -1.03% | -0.80% |
| 5 | 0.00% | 0.00% | -4.69% | -4.47% | -2.56% | -0.72% |
| 7 | 0.00% | 0.00% | -7.33% | -6.73% | -3.21% | -3.35% |
| 9 | 0.00% | 0.00% | -8.12% | -8.46% | -6.22% | -4.20% |

(a) Index size.

| # Workers | $k$-mer % $\Delta$ | | Color Classes % $\Delta$ | |
|---|---|---|---|---|
| | RR | WR | RR | WR |
| 3 | -7.57% | -7.13% | -2.48% | -1.93% |
| 5 | -8.64% | -8.74% | -2.95% | -3.05% |
| 7 | -13.10% | -9.06% | -6.61% | -5.10% |
| 9 | -14.78% | -11.22% | -8.12% | -5.81% |

(b) Index representation.

**Table 5: Percent change in Mantis representation and index size for similarity-based load-balancing relative to Round-robin (RR) and weighted-random (WR) baselines for distributing 1,500 experiments as the number of simulated workers increases.**

can see the relative improvement of similarity-based load-balancing also increases with worker count, for a fixed number of experiments. For example, total $k$-mer count reduction relative to Round-robin increases from $\approx 7.6\%$ using 3 workers to $\approx 14.8\%$ with 9 workers. This is likely explained by increasing worker counts allowing for more fine-grained subdivisions of the experiment space, resulting in more compact clusters.

## 5 DISCUSSION

In this paper, we presented a proof-of-concept implementation of a similarity-based load-balancer for distributed genomic search indices, evaluated on the Mantis system. Our initial results indicate that similarity-based load-balancing produces more compact color class tables and smaller overall Mantis indices than naive methods and further that these gains seem to increase with scale, suggesting that structure-aware file distribution is a viable general strategy for distributed $k$-mer indexing at scale.

Although evaluated only on a small subset of the SRA, the preliminary results are promising and motivate scaling the approach to larger and more diverse datasets. There are a number of exciting directions for future work. From a computational standpoint, we plan to conduct more in-depth scaling studies analyzing additional dimensions including larger $k$-mer sizes and increased SRA diversity, and more closely characterize the effect of load-balancing method on worker load variance, query time and index update speed. Prior work [31] has also shown that clustering similar SRA experiments before indexing can reduce index size, suggesting additional optimization opportunities. While the current implementation is primarily a prototype, substantial performance improvements remain possible through logarithmic-time query structures, redesigns tailored to distributed environments and network-aware data movement. Additionally more complex concurrent sketch construction methods leveraging multiple-threads to process a single file, may help alleviate the I/O bottleneck we are currently observing.

From a biomedical application perspective, we are interested in applying distributed, updatable genomic search indices with similarity-based load-balancing to data portals such as the NCI Genomics Data Commons [11] or NHLBI BioData Catalyst [18]. Furthermore, we are interested in integrating Mantis indices with other semantic tools such as knowledge graphs to create unified interoperability over portals, which could enable the automated interpretation of retrieved genomic experiments within the context of their studies as well as create a path towards multi-modal data integration. Additionally, we plan to investigate if the performance gains we observed on the Breast, Blood, and Brain benchmark dataset hold for more biologically diverse datasets integrating multiple species, tissues, experimental designs and assay types. Finally we would like to investigate if biological motivation and domain knowledge we are not currently considering could further improve load-balancing performance.

## AUTHORS

**Woodward (Buz) Galbraith**, PhD Student / Northeastern University
*Works on large-scale data integration with knowledge graphs for biomedical applications in the Gyori Lab.* Community: biomedical and data management.

**Ilgar Mammadov**, PhD Student / Northeastern University
*Works on network security and privacy in Internet of Things systems, under the guidance of Prof. David Choffnes and Dr. Daniel J. Dubois.* Community: data management.

**Shreyaan Pathak**, Undergraduate Student / Northeastern University
*Works on AI systems approaches, specifically methods for efficient Multi-GPU checkpointing for LLM training with Prof. Gene Cooperman.* Community: data management.

**Benjamin M. Gyori**, Associate professor / Northeastern University
*Works on large-scale data integration and modeling in biomedicine. His research combines computational systems modeling, ML, NLP, and human–machine interaction to improve our understanding of complex human biology.* Community: biomedical and data management.

**Prashant Pandey**, Assistant Professor / Northeastern University
*Develops scalable data systems and probabilistic data structures (e.g., filters, sequence*

*indices) with strong theoretical guarantees. His work has direct impact on downstream applications like high-performance computing and computational biology.* Community: data management.

## ACKNOWLEDGMENTS

This work was partially supported under the DARPA ASKEM and ARPA-H BDF programs (HR00112220036).

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
