# OpenReview forum: "Similarity-Based Load-Balancing for Distributed Genomic Indices"
_VLDB.org/2026/Workshop/BioDMS — BioDMS 2026 ProjectTalk_

### Official Review · Reviewer_Dh5F · 2026-06-12

**Summary:**

This paper investigates the problem of efficient search over large-scale genomic sequencing data and addresses a key limitation of existing work, namely the assignment of newly added experiments to nodes, which can lead to redundant storage and inflated index sizes. To address this issue, the authors propose a similarity-based load-balancing strategy that can be integrated into a broad class of updatable distributed genomic search indexes. Building upon this strategy, the paper presents a concrete system implementation that leverages genomic sketching techniques to distribute files across nodes. The experimental evaluation, conducted on a 750-experiment subset of the Breast, Blood, and Brain dataset from the NIH Sequence Read Archive (SRA), demonstrates the effectiveness of the proposal from multiple perspectives.

**Confidence Of Review:**

3

**Detailed Feedback Points:**

**Strong Points:**

* S1. The paper addresses an important and promising problem, namely enabling similarity-based load balancing for distributed genomic indices to support space-efficient incremental updates. The motivation is clearly presented, and the proposed direction has the potential to improve the scalability and efficiency of search over large-scale genomic sequencing data.

* S2. The proposed similarity-based load-balancing strategy is technically sound and broadly applicable to updatable distributed genomic search indices. The accompanying system implementation, which leverages genomic sketching techniques, is well aligned with the stated objectives and demonstrates a coherent system design.

* S3. The experimental evaluation compares the proposed similarity-based load-balancing strategy against representative baselines across multiple aspects, including index size, index representation, and runtime performance. The results provide evidence of the effectiveness and efficiency of the proposed strategy under the evaluated settings. Furthermore, the authors have released the source code, which improves the reproducibility of the work.

**Opportunity for Improvement:**

* O1. Section 5 identifies the integration of Mantis indices with semantic tools such as knowledge graphs as a promising future research direction. It would strengthen the paper to further elaborate on the types of biomedical domain knowledge that could be incorporated into the proposal and to discuss the potential benefits such integration could bring to genomic search and indexing.

* O2. Related to the above point, it would be valuable to discuss how insights derived from downstream biomedical applications could, in turn, inform the design of the proposed load-balancing strategy and system architecture. Such a discussion could help clarify the broader impact of the work and highlight opportunities for future co-design between domain knowledge and systems development.

**Relevance For Biodms:**

3

---

### Official Review · Reviewer_wkJ6 · 2026-06-17

**Summary:**

This paper evaluates a solution for a practical problem in searching large sequencing repositories. That is, as new experiments are added to a distributed search index, similar samples often end up on different nodes, which duplicates data and causes bloat. The authors propose routing each new experiment to whichever node already holds the most similar data and show on a 750-experiment RNA-seq subset that this produces smaller, more evenly balanced indices than the current approach. The results are promising but preliminary, and it remains to be seen whether the benefits hold on larger and more biologically diverse data.

**Confidence Of Review:**

2

**Detailed Feedback Points:**

- The study design isolates the source of the improvement nicely. By comparing against a random baseline that assigns the same number of files per worker, the authors show the gains come specifically from grouping genomically similar experiments, and not simply from an uneven file distribution.
- The evaluation is narrow, both in scale and in biological diversity. The results come from 750 RNA-seq experiments drawn from only three tissue types (breast, blood, and brain), with modest improvements. It is not clear the same benefit would hold across the far more heterogeneous data in the full SRA, where samples span many species, tissues, assays, and study designs. More discussion around this point would be interesting and informative.
- The systems details are outside my area of expertise, but it would be valuable to see how this behaves at query time rather than only during index construction. Grouping similar experiments may speed up some searches, but it could also concentrate load if researchers tend to query within a single tissue or disease area, since related data would all sit on the same node.
- I am also interested in discussion around whether this could optimize processing speed/times for large genomics cohorts.

**Relevance For Biodms:**

3

---

### Official Review · Reviewer_TLWy · 2026-06-20

**Summary:**

This paper addresses a highly relevant problem for distributed genomic indexing systems. The proposal to use sketching methods to guide load-balancing based on genomic similarity is clever and potentially very impactful. The preliminary results, showing reductions in total k-mer count and variance across workers, are promising.

**Confidence Of Review:**

3

**Detailed Feedback Points:**

Strong Pts:
- The problem of k-mer redundancy in distributed indices is clearly explained, and the proposed solution has been correctly motivated.
- The fact that the balancing strategy is independent of the underlying index (like Mantis) increases its potential impact and generalizability. The inclusion of a weighted-random load-balancing control is a methodological strength.

Comments:
- The bottleneck represented by sketch construction time is extremely significant and can affect system efficiency.

- Although the authors acknowledge this is a prototype limitation and plan improvements, it remains a critical issue.

- In a real-world scenario with millions of experiments, the cost of computing similarity could easily outweigh the benefits gained in index compactness.

- Beyond build time, have the authors evaluated the impact of adding a new experiment to a similar worker? In particular, I suggest measuring the impact on the update time of the CQF and the MST tree when experiments are grouped by similarity, w.r.t. random assignment.

- Could the authors also provide more details on the experimental setup, such as the number of nodes used in their simulated distributed system?

**Relevance For Biodms:**

3